# The intrinsically disordered N-terminus of the voltage-dependent anion channel

**Jordane Preto**[1]*, **Isabelle Krimm**[1,2]

**1** Université Claude Bernard Lyon 1, Centre de Recherche en Cancérologie de Lyon, Centre Léon Bérard, INSERM 1052, CNRS 5286, Lyon, France, **2** CRMN, UMR CNRS 5082, ENS de Lyon, Université Lyon 1, Villeurbanne, France

* jordane.preto@univ-lyon1.fr

**Data Availability Statement:** All original data files and scripts are available from the Dryad Digital Repository: https://doi.org/10.5061/dryad.zw3r2286m.

## Abstract

The voltage-dependent anion channel (VDAC) is a critical β-barrel membrane protein of the mitochondrial outer membrane, which regulates the transport of ions and ATP between mitochondria and the cytoplasm. In addition, VDAC plays a central role in the control of apoptosis and is therefore of great interest in both cancer and neurodegenerative diseases. Although not fully understood, it is presumed that the gating mechanism of VDAC is governed by its N-terminal region which, in the open state of the channel, exhibits an α-helical structure positioned midway inside the pore and strongly interacting with the β-barrel wall. In the present work, we performed molecular simulations with a recently developed force field for disordered systems to shed new light on known experimental results, showing that the N-terminus of VDAC is an intrinsically disordered region (IDR). First, simulation of the N-terminal segment as a free peptide highlighted its disordered nature and the importance of using an IDR-specific force field to properly sample its conformational landscape. Secondly, accelerated dynamics simulation of a double cysteine VDAC mutant under applied voltage revealed metastable low conducting states of the channel representative of closed states observed experimentally. Related structures were characterized by partial unfolding and rearrangement of the N-terminal tail, that led to steric hindrance of the pore. Our results indicate that the disordered properties of the N-terminus are crucial to properly account for the gating mechanism of VDAC.

## Author summary

The voltage-dependent anion channel (VDAC) is a membrane protein playing a pivotal role in the transport of ions or ATP across the mitochondrial outer membrane as well as in the induction of apoptosis. At high enough membrane potential, VDAC is known to transition from an open state to multiple closed states, reducing the flow of ions through the channel and blocking the passage of large metabolites. While the structure of the open state was resolved more than a decade ago, a molecular description of the gating mechanism of the channel is still missing. Here we show that the N-terminus of VDAC is an intrinsically disordered region and that such a property has a profound impact on its dynamics either as a free peptide or as part of the channel. By taking disordered properties

**Funding:** This work was supported by the Ligue contre le cancer, Comité de la Haute-Savoie (IK). The funders had no role in study design, data collection and analysis, decision to publish, or preparation of the manuscript.

**Competing interests:** The authors have declared that no competing interests exist.

of the N-terminus into account, we managed to generate long-lived closed conformations of the channel at experimental values of the membrane potential. Our results provide new insights into the molecular mechanism driving the gating of VDAC.

## Introduction

The voltage-dependent anion channel (VDAC) is a major porin located in the mitochondrial outer membrane (MOM). One of VDAC's main function is to mediate the transport of ions or small metabolites such as ATP, ADP or NADH between the mitochondrial intermembrane space (IMS) and the cytoplasm thus contributing to cell homeostasis [1]. In addition, the channel is responsible for alteration of the permeability of the MOM which may result, under specific stimuli, to the release of apoptogenic factors like cytochrome C from the IMS, eventually leading to apoptosis [2]. Through its ability to interact with a dozen different proteins – including itself – VDAC was shown to either induce or inhibit apoptosis depending on its interaction partner [2]. For this reason, VDAC is usually considered a promising target not only for cancer treatment but also in neurodegenerative diseases where the main goal is to prevent early stages of neuron's death [3].

Single-channel conductance experiments have revealed that VDAC can either be found in an open, high conducting, state prevailing at low membrane potential (between −30 mV and 30 mV) or in multiple closed, low conducting, states occurring at more extreme voltage (above or below ±30 mV) [4, 5]. Importantly, anion selectivity can be fairly high in the open state with an anion/cation ratio of 5–6 at 150-mM KCl concentration while closed states were reported to be cation selective [5, 6]. Although the structure of the open state of VDAC was resolved more than a decade ago, little is known regarding the structure of the closed states due to their transient properties in normal conditions [4].

Experimental structures of the open state of VDAC were first released in 2008 by three separate groups using X-ray crystallography and/or NMR [7–9]. VDAC is composed of 19 β-strands forming the barrel as well as an α-helical N-terminus (residues M1 to G25) located midway inside the pore and interacting with the barrel mostly via hydrophobic contacts (Fig 1). While the structure of the β-barrel was similar in all cases, slight differences in the position and in the α-helical content of the N-terminus, i.e., long helix vs. small helix vs. helix-break-helix, were observed between published structures [10]. Although the N-terminus was reported to be stable in the open state (no large conformational change on the sub-millisecond timescale [11]), these slight variations may be an indicator of local flexibility of that region on longer timescales or different arrangements due to external conditions such as lipid environment [10]. The mobility of the N-terminus was confirmed in a wide range of biophysical studies suggesting that the segment could detach from the pore and even translocate outside the channel during voltage gating [12]. Nevertheless, N-terminal translocation does not seem to be required for gating as several experimental studies introducing a disulfide bridge to "lock" the N-terminus against the β-barrel wall showed that closed states may still arise when the segment is located inside the pore [13, 14]. Such studies include that of Teijido and al. [14] where the introduction of an S-S bond at L10C-A170C was found to have no effect on the conductance profile of the channel confirming that closure can still be induced without any large conformational change of the N-terminus.

Finally, important structural properties of the N-terminus of VDAC were discovered by De Pinto and coworkers [15, 16]. Using NMR and circular dichroism (CD) experiments, the authors found that the N-terminal fragment, when produced as a free peptide in solution, was

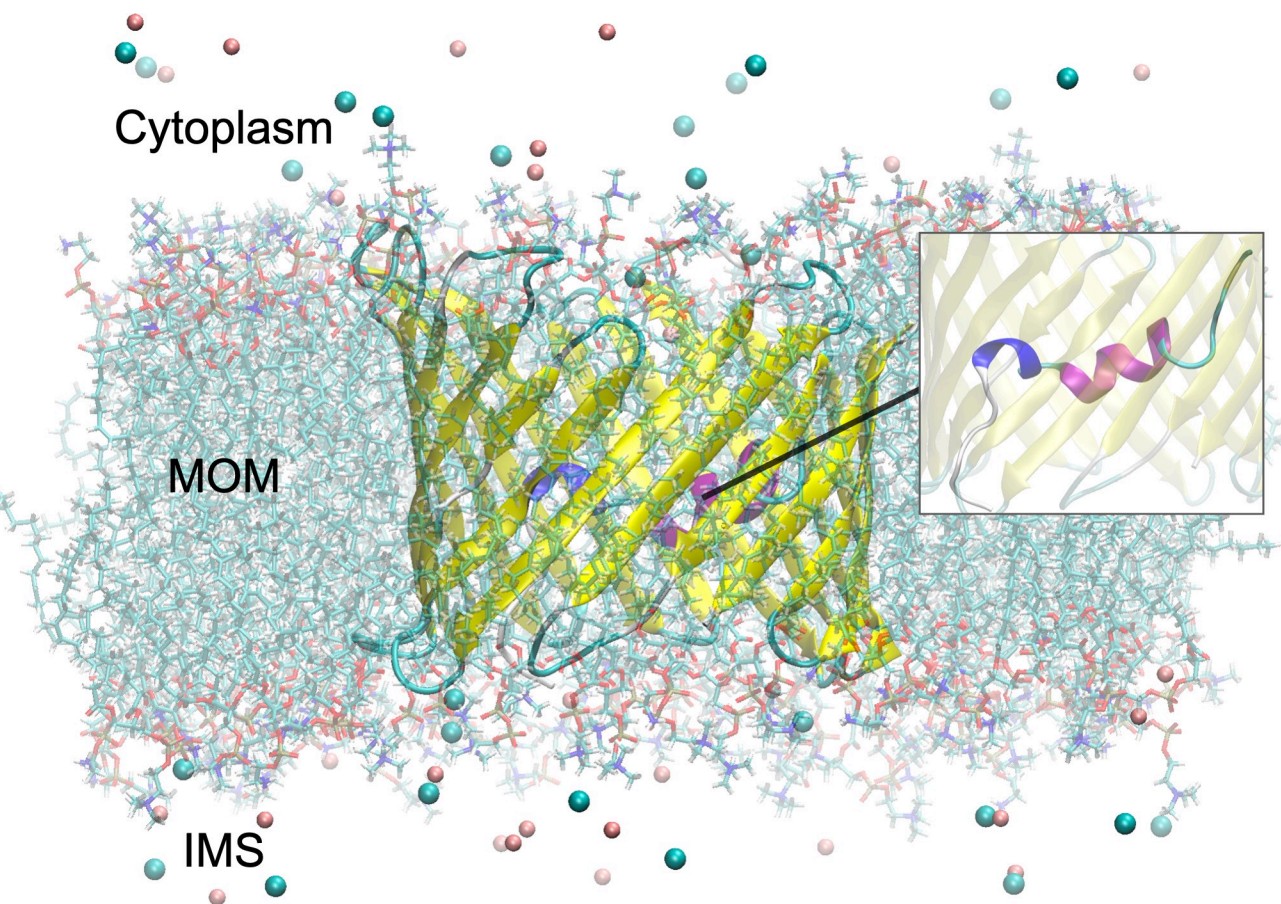

**Fig 1. VDAC embedded in a lipid bilayer.** The structure of the murine VDAC1 (mVDAC1) solved by X-ray crystallography was used (PDB ID: 3EMN [9]). The inset shows a closer view of the N-terminal region with the $3_{10}$-helix involving Y7, A8 and D9, depicted in dark blue, and the standard helix (K12 to T19) in fuchsia. Chloride (in blue-green) and potassium ions (in pink) are also shown at a 150-mM KCl concentration for illustrative purposes. The MOM as well as expected cytoplasm and IMS sides are also indicated.

characterized by no apparent structural content. From molecular dynamics (MD) simulations, it was proposed that such a peptide could exist in a dynamic equilibrium alternating between random coil and α-helix content while interaction of the N-terminal region with the inner part of the VDAC barrel would "choose" the α-helix conformation over unstructured ones [16].

In the present paper, it is shown that the lack of apparent structure experimentally observed by De Pinto and coworkers is a simple illustration of the disordered nature of the N-terminus which, in our opinion, has not properly been investigated until now. By running MD simulations with a force field specifically designed for intrinsically disordered regions (IDR), we showed that the conformational landscape of the N-terminal peptide is mostly unstructured in solution with almost no helical content. Alternatively, simulation with a standard force field similar to what has been used in previous in silico studies, was found to overstabilize its α-helix content leading to erroneous results. Our findings were confirmed by the ability of the IDR force field to reproduce experimental Cα and Cβ chemical shifts both qualitatively and quantitatively, in contrast to the standard force field.

In the second part, we explored the importance of the disordered properties of the N-terminus in the gating mechanism of VDAC. Specifically, experiments by Teijido et al. [14] were

modeled by introducing a disulfide bond between residues L10C and A170C. Using the IDR-specific force field, accelerated dynamics trajectories at ±40 mV voltage were characterized by the unfolding of the $3_{10}$-helix involving residues Y7, A8 and D9 and subsequent rearrangements of disordered residues M1 to D9. At +40 mV, the unfolded segment was found to unwind and to move towards the middle of the pore leading to steric hindrance of the channel. Metastable states displaying a low conductance representative of closed states (~50–60% of the conductance in the open state) as well as a reduced anion selectivity, were observed. Specific positioning of the unfolded N-terminal segment in our metastable states as well as the inability for standard force fields to induce stable low-conducting states suggest that the disordered properties of the N-terminus are critical to account for the closing mechanism of VDAC. Although our subconducting states were generated at +40mV, the exact role of the membrane potential in channel closure, as compared with random voltage-independent events, is still to be determined.

## Results

### The N-terminal peptide

We first investigated the disordered properties of the N-terminal peptide in solution. Starting from the α-helical N-terminus (M1-G25) directly extracted from the crystal structure of mVDAC1 [9] (PDB ID: 3EMN), we performed μs-long MD simulations using, on one side, the Amber ff14SB force field which is considered one of the standard and latest benchmarks for protein simulations [17] and, on the other side, the ff14IDPSFF force field that was specifically designed to sample disordered protein domains [18]. Importantly, ff14IDPSFF was reported to produce results similar to ff14SB in the case of well-folded systems [18].

Representative conformers of the N-terminal peptide obtained after running standard MD with the ff14SB and ff14IDPSFF force fields are shown in Fig 2 (top). Long MD trajectories of 1.6 μs and 2.1 μs were run with ff14SB and ff14IDPSFF, respectively, while convergence was confirmed in both cases as the number of clusters generated from clustering analysis was found to plateau in the last 500 ns (S1 Fig). Notably, important differences in the conformational landscapes of the two force fields were observed. Whereas dynamics performed with ff14SB was mostly characterized by α-helices involving residues Y7 to K20, simulation with the ff14IDPSFF force field led to unstructured conformations including partially as well as fully unfolded structures. To get further insight into the stability of each state, the potential of mean force (PMF) was provided (S2 Fig) as a function of the RMSD with respect to the crystal structure and the radius of gyration ($R_g$), two popular collective variables for structural analysis of small peptides [19]. As depicted in Figs 2 and S2, transient secondary structures were also identified using ff14IDPSFF including small antiparallel β-strands made of A2-V3 and Y22-G23 (RMSD~*11.5 Å*, $R_g$~*8.5 Å*) as well as an α-turn involving D9, L10 and G11 (RMSD~*9 Å*, $R_g$~*9 Å*). The latter was found to minimally contribute to the overall conformational ensemble (1.24%).

Although our simulations were μs-long and convergence was reached in all cases, we explored the possibility that dynamics carried out with ff14SB could produce other conformations of the peptide (over longer timescales) with no α-helical content. To this purpose, we used accelerated Molecular Dynamics (aMD), a biasing technique originally developed by the McCammon group [21]. aMD has been successfully applied to a vast diversity of systems including the 58-residue pancreatic trypsin inhibitor whereby 500-ns-long aMD simulation allowed to sample conformations normally reachable on the millisecond timescale [22]. Hence, aMD turns out to be a method of choice to investigate the conformational landscape of our 25-residue peptide more thoroughly.

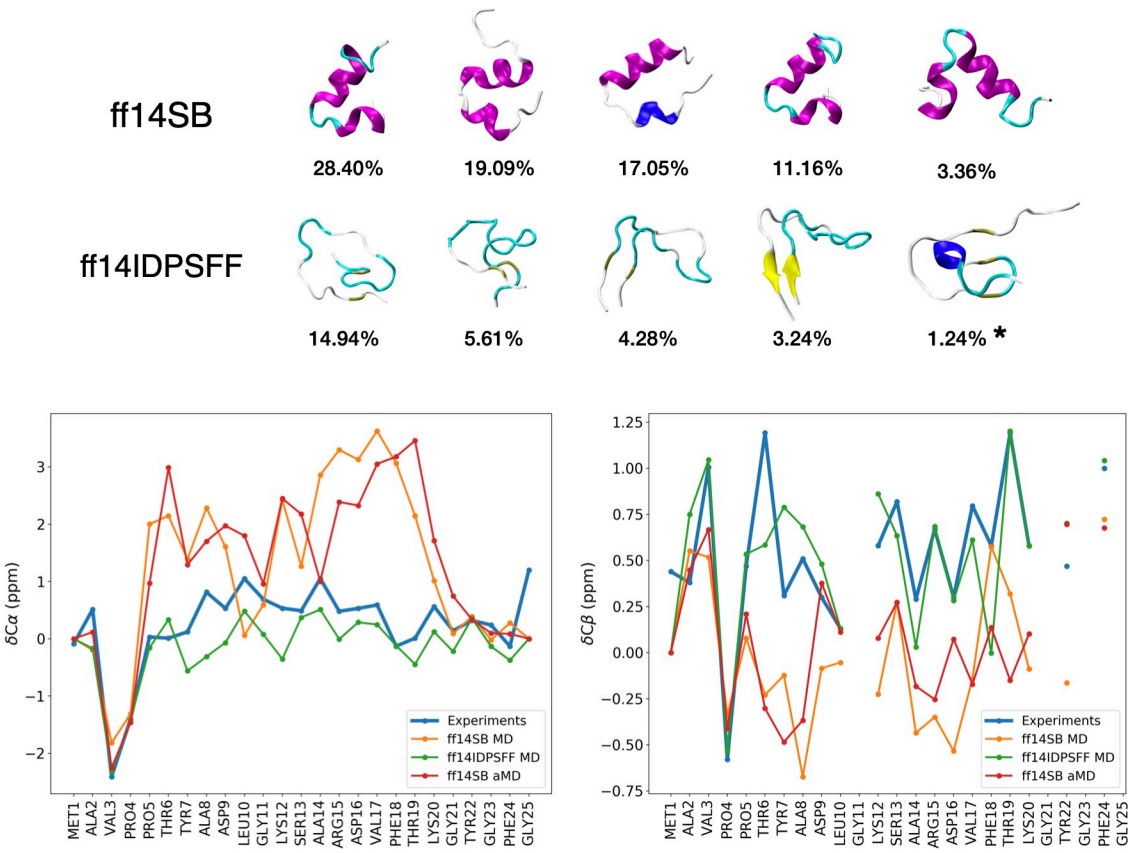

**Fig 2. Representative conformers and chemical shifts of the VDAC1 N-terminal peptide.** Top: representative conformers of the top 5 clusters (and their occupancy) obtained from μs-long simulations with ff14SB and ff14IDPSFF. Note that the last (starred) structure shown for ff14IDPSFF corresponds to the 12th most populated cluster which is the first cluster with α-helical content. Bottom: α- and β-carbon chemical shifts of the N-terminal peptide predicted from molecular simulations and experiments (experimental values were directly extracted from Table S1 in Ref. [16]). Shifts computed from MD simulations were obtained by simple average of the shifts predicted for each generated structure using the SPARTA+ software [20]. Shifts computed from aMD simulations were calculated as weighted averages using Maclaurin series up to rank k = 10 in order to approximate the weights (Eq (4)).

Overall, the conformational landscape generated from a 1μs-long aMD simulation with ff14SB was similar to the α-helix ensemble generated from standard MD using the same force field, with no new basin or new structural content reported (S3 Fig). To eliminate the possibility that our simulations could have been biased by our choice of starting structure, i.e., the α-helical N-terminus extracted from mVDAC1, which may have led to the peptide being "trapped" in an α-helix minimum, we also ran a short aMD simulation (20 ns) starting from a fully unfolded structure ($R_g$~20 Å). This time was short enough to observe clear convergence to the main α-helix basin found in our long simulation suggesting that further stable or metastable regions with ff14SB were unlikely (S3 Fig).

Finally, we investigated the correctness of our results by estimating the NMR chemical shifts of alpha and beta carbons from each simulation and by comparing those values with experimental ones reported elsewhere [16]. Results are provided in Fig 2 (bottom) for both ff14SB and ff14IDPSFF force fields where the former is given in cases of standard MD and aMD simulations. Notably, Cα and Cβ chemical shifts predicted using ff14SB were found to be very far from experimental data (chemical shifts obtained from MD and aMD simulations were similar, which is consistent with the similar landscapes observed). Discrepancies were

especially visible for α-helix residues confirming the tendency of ff14SB to overstabilize secondary structures in disordered systems [23]. Conversely, both Cα and Cβ shifts computed from the ff14IDPSFF trajectory showed great agreement with experiments. From S1 Table, we can see that the improvement was both quantitative and qualitative as low RMSE values and high Pearson correlation coefficients (0.870 for Cα and 0.759 for Cβ) were obtained, respectively.

## The N-terminus in VDAC

Here we explored the stability of the intrinsically disordered N-terminus as part of the open state of VDAC. MD simulations of mVDAC1 were performed at physiological salt concentration (150 mM KCl) starting from the 3EMN structure [9] embedded in a DOPC/DOPE membrane (Fig 1). As with the N-terminal peptide, standard MD was performed with ff14IDPSFF and compared to simulations using the ff14SB force field. Since ff14IDPSFF has not yet been tested on proteins interacting with lipids, the new force field parameters were only applied to the N-terminal region which is located inside the pore, while ff14SB was always used to model the rest of the channel.

In Fig 3, we showed the root mean square fluctuations (RMSF) of N-terminal residues over 100 ns while the inset displays the N-terminal RMSD as a function of time. At first glance, we noticed that both force fields resulted in almost equally-rigid N-terminal regions (RMSD < 2 Å) although many residues including L10 to D16 and Y22 to G25 had a slightly higher RMSF in the case of ff14IDPSFF. Interestingly, this increase in flexibility was not necessarily correlated with the ability of secondary structures to unfold. For example, the secondary structure propensity obtained with ff14IDPSFF in the long α-helix (K12 to T19) was equal or even higher than the propensity calculated with ff14SB (S4 Fig). On the opposite, the $3_{10}$-helix involving Y7, A8 and D9 which is characteristic of the 3EMN structure, was slightly less stable in the case of ff14IDPSFF (0.86 and 0.76 propensity for ff14SB and ff14IDPSFF, respectively).

Overall, we found that the ff14IDPSFF force field predicted a stable and well-folded N-terminal region within the timeframe of our standard MD run (650 ns, see S2 Table) with no apparent transition or change in structural content compared to the 3EMN structure (S1 Video). Not only is this result consistent with experiments showing the stability of the N-terminus in the open state of VDAC [24] but it also confirms the ability of ff14IDPSFF to model stable IDRs as they fold upon interaction with other protein domains.

## Double cysteine mutant

To investigate the closing mechanism of VDAC, simulations of a double cysteine mutant (mVDAC1-Cys) characterized by a disulfide bond connecting residues L10C and A170C, were performed. Electrophysiological measurements by Teijido et al. [14] have revealed that this modification was not altering the gating properties of the channel even though the N-terminus was affixed against the barrel. Therefore, such a setup provides a good way to investigate closure of VDAC from molecular simulations as the presence of the S-S bond should dramatically restrict the exploration of the conformational space of the N-terminus, thereby saving a lot of computational time. In addition, our results should confirm that closure of the channel is not necessarily induced by large conformational changes of that region.

All our simulations were initiated from the 3EMN structure where we mutated residues L10 and A170 to create the disulfide bond. Since L10 and A170 were already spatially close to each other in 3EMN (min. interatomic distance ~ 3.9 Å), the mutated channel was found to relax to a state similar to the crystal structure (Fig 4). As with mVDAC1 WT, standard MD runs – using either ff14SB or ff14IDPSFF – were unsuccessful in simulating conformational

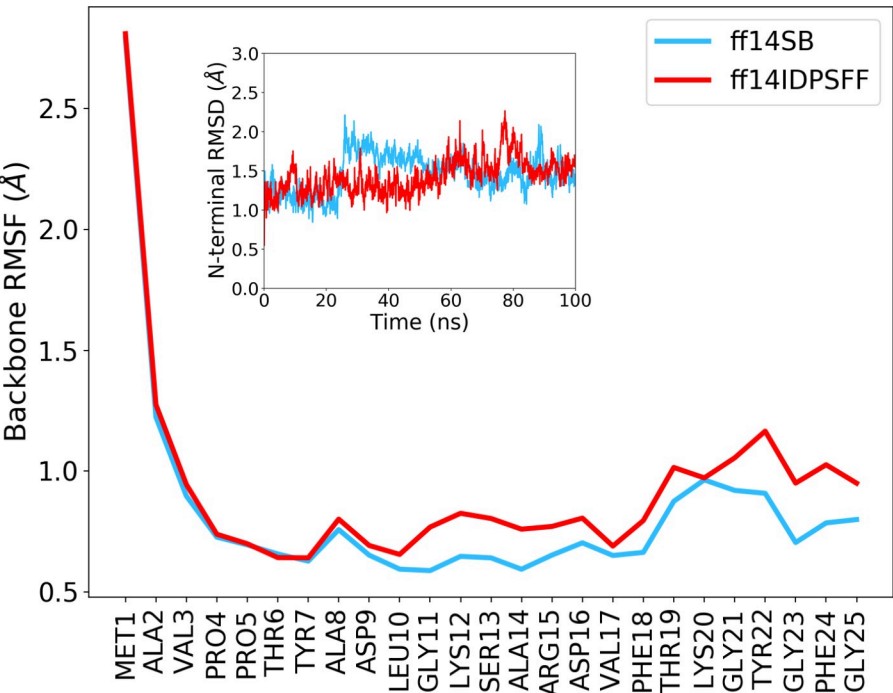

**Fig 3. RMSF and RMSD of the N-terminus of wild-type mVDAC1.** RMSF and RMSD were computed from 100-ns-long runs using ff14SB or ff14IDPSFF. Only the backbone atoms of the N-terminus were considered for the calculation of RMSF. Inset: N-terminal RMSD with respect to the 3EMN structure as a function of time. RMSD values were calculated by first aligning the structures over all the heavy atoms including atoms of the pore and of the N-terminus.

transitions of the N-terminus or the barrel. Therefore, aMD simulations were carried out. Three membrane potentials: 0 mV, +40mV and −40 mV, at 150 mM KCl, were considered (one aMD trajectory per potential). Voltage was defined as positive when it was greater at the

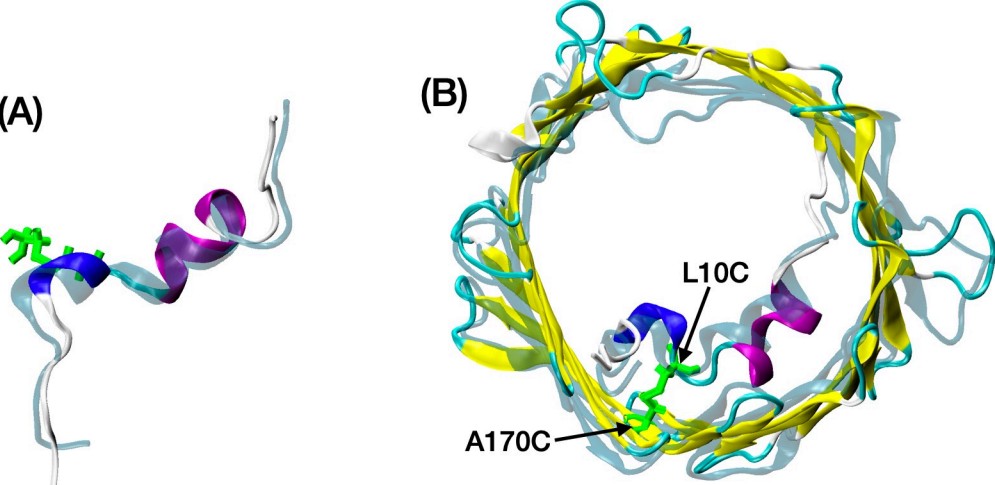

**Fig 4. mVDAC1-Cys structure relaxed from 3EMN.** mVDAC1-Cys conformation obtained after 10-ns-long equilibration of the (mutated) 3EMN crystal structure using ff14IDPSFF. The crystal structure is depicted in transparent turquoise. Disulfide bridge linked L10C and A170C residues are shown in green. (A) N-terminal region (side view); (B) whole channel (IMS view; the IMS is expected to correspond to the *trans* side in PLB experiments).

side opposite to the ends of N- and C-termini in the open state. Importantly, this side was found to correlate with the cytosolic side of the channel that was also shown to correspond to the *cis* side in planar lipid bilayer (PLB) experiments [25, 26]. Therefore, the potential values specified in the present work most likely represent potential values set at the *cis* side in PLB experiments while setting the *trans* side to 0 mV.

aMD simulations were conducted over hundreds of nanoseconds at each voltage (a summary of our simulations including simulation times, is given in S2 Table). While trajectories generated with ff14SB revealed no major displacements or structural changes of the N-terminus at 0 mV or at applied potential (S2 Video), simulations with ff14IDPSFF showed more complex dynamics depending on the voltage: aMD trajectories at +40 mV and −40 mV were both characterized by early unfolding of the $3_{10}$-helix involving residues Y7, A8 and D9 (after 20 ns at +40 mV and after 2 ns at −40 mV). At 0 mV, the $3_{10}$-helix was more stable although our aMD trajectory also led to unfolding of this region after 160 ns (S3 Video) suggesting that such an event may not be fully induced by the membrane potential. Despite unfolding of the helix, position and orientation of the N-terminus including those of the N-terminal tail (M1 to T6) remained globally unchanged at 0 mV (S5 Fig) as evidenced by hydrogen bonds between M1 and E121, A2 and H122, or P4 and N124 prevailing all along our simulation.

Contrary to the 0 mV case, our +40 mV aMD trajectory showed a detachment of the N-terminal tail from the barrel whereby residues M1 to P5 moved towards the middle of the pore around 140 ns (S4 Video). At this stage, the tail was found to either interact with the β-barrel at strands 4 and 5 (hydrogen bonds at M1-E84, A2-E84), move almost freely or interact with other residues of the N-terminus (hydrogen bonds at M1-D9 and A2-Y7) indicative of a high flexibility of that region (S5 Fig). At −40 mV, we noticed that the N-terminal tail remained in a more "packed-up" conformation (S5 Video and S5 Fig). Although residue M1 was found to interact electrostatically with E121 just as in the 0 mV case, hydrogen bonds and hydrophobic contacts between residues T6, Y7, A8 and strands 7 and 8 were also reported, which led to slight apparent distortion of the barrel in the region of interaction (Cluster #1 in S5 Fig). Remarkably, while the $3_{10}$-helix experienced unfolding and subsequent repositioning at different voltages, the long α-helix made of residues K12 to T19 always remained stable and well-folded with a position similar to that of the crystal structure (S5 Fig).

To investigate whether any of the conformations generated from aMD could correspond to closed states of VDAC, we performed grand canonical Monte Carlo/Brownian dynamics (GCMC/BD) which enable to rapidly simulate the flow of ions across membrane channels [27] GCMC/BD has been successfully applied to porin systems [28] including the open state of VDAC [29]. In GCMC/BD, only ions are explicitly simulated while the rest of the system including the protein and the membrane, is modeled as a fixed continuous dielectric. In the present study, clustering analysis was carried out on each aMD trajectory (0 mV, +40 mV and −40 mV) by generating 500 clusters in each case. Next, independent GCMC/BD trajectories were run at 150-mM KCl from the representative structure of each cluster to approximate the conductance and anion/cation ratio of each state. Estimates were obtained by running GCMC/BD at a single voltage value for each frame (at +40 mV for frames generated at +40 mV and 0 mV, and at −40 mV for frames extracted from the −40 mV trajectory).

From this approach, we observed that the conductance of the channel remained globally stable at 0 mV with an average value of 0.918 nS and few conformations below 0.6 nS (S6 Fig). Importantly, this mean value was relatively close to the experimental conductance value of the open state of 0.81 nS at 150-mM KCl concentration [6]. As another remark, no visible change in the conductance was reported after unfolding of the $3_{10}$-helix (160 ns) confirming that the channel was still in its open state after such an event. Regarding the anion/cation ratio, experiments have estimated it to be around 5–6 in the open state of VDAC1 WT at 150 mM KCl [6].

In our simulation, large fluctuations of this ratio, characterized by high values (>40) and an average value of 14, were reported. We speculate that such high ratios are due to the small number of cations crossing the pore during our GCMC/BD runs (often less than 5), which are expected to lead to an overestimation of the anion selectivity in the open state.

At +40 mV, our GCBC/BD runs revealed a drop in the conductance at about 250 ns in our aMD simulation with many structures below 0.6 nS (Fig 5), where 0.6 nS corresponds to 65.3% of the conductance measured at 0 mV. At this stage, the N-terminus was characterized by a zigzag unfolded segment (from M1 to D9) located midway inside the pore and interacting with the β-barrel mostly via hydrophobic contacts (Fig 5 and Cluster #5 in S5 Fig). The drop in the conductance was coupled by a significant decrease in the anion/cation ratio with values less than 3 as well as a few structures exhibiting slight cation selectivity. From 280 ns, conductance slowly increased again which happened when the tail detached again from the barrel. Unlike the +40 mV case, our GCBC/BD simulations performed on our −40 mV aMD trajectory did not show any long-lived subconducting state although a few transient conformations characterized by both a low conductance and reduced anion selectivity or even cation selectivity were observed (S6 Fig).

## Closed state(s)

To investigate the stability of the subconducting conformers generated from aMD at +40mV, standard MD simulations were performed. Out of the 500 representative structures tested using GCMC/BD, 51 frames were predicted to have a conductance less than 0.6 nS between 240 ns and 290 ns (red stars in the dashed rectangle in Fig 5 (top)) and were therefore used as starting point for 200-ns-long MD runs at +40mV (one trajectory per conformer). For each trajectory, the electric current was estimated by monitoring the total number of ions crossing the channel and by estimating the slope of the *crossing-events-vs-time* curve from linear regression. As is well known, standard MD tends to overestimate the diffusion coefficient of ions in explicit solvent [29]. Hence, a multiplicative factor based on both experimental and simulated bulk diffusion coefficients of potassium chloride was used to compute the current (see production MD (channel) in material and methods). As a control, our approach was applied to a 450-ns-long MD trajectory of our initial mVDAC1-Cys structure equilibrated from 3EMN (Fig 4). The predicted current was 34.98 pA at +40mV resulting in an (approximated) conductance of 0.87 nS which is well in line with the experimental value of 0.81 nS reported in the open state. At the same time, the anion/cation ratio was found to be 5.96 at the end of the simulation which is also quite consistent with the 5–6 ratio observed in experiments.

Regarding our subconducting conformers, a summary of the results is provided in S3 Table. Among the 51 structures, five were shown to exhibit both a low conductance (around 0.6 nS or less) and a reduced anion/cation ratio (around 4 or less). With a pairwise RMSD of 1.76 Å, two of the five conformers were found to correspond to the same structural state now referred to as S1 state, while the remaining three also belonged to the same S2 state (see S7 Fig). Since the S1 state displayed a lower apparent conductance (~ 0.5 nS) and anion/cation ratio (< 3) than S2, we chose to investigate the former more thoroughly by extending the trajectory of one such conformer up to 450 ns at +40mV and by comparing the results with those of the open state. The number of crossing events as well as the anion/cation ratio are shown in Fig 6 for the open and S1 states. In both cases, we noticed that the number of crossing events described an almost straight line as a function of time confirming the stability of each state in terms of conductivity. The ratio of crossing events between S1 and the open state was 96/167 ~57.5%, at the end of the simulation which falls within the range of conductance ratios reported experimentally and is consistent with our previous GCMC/BD runs. The anion/

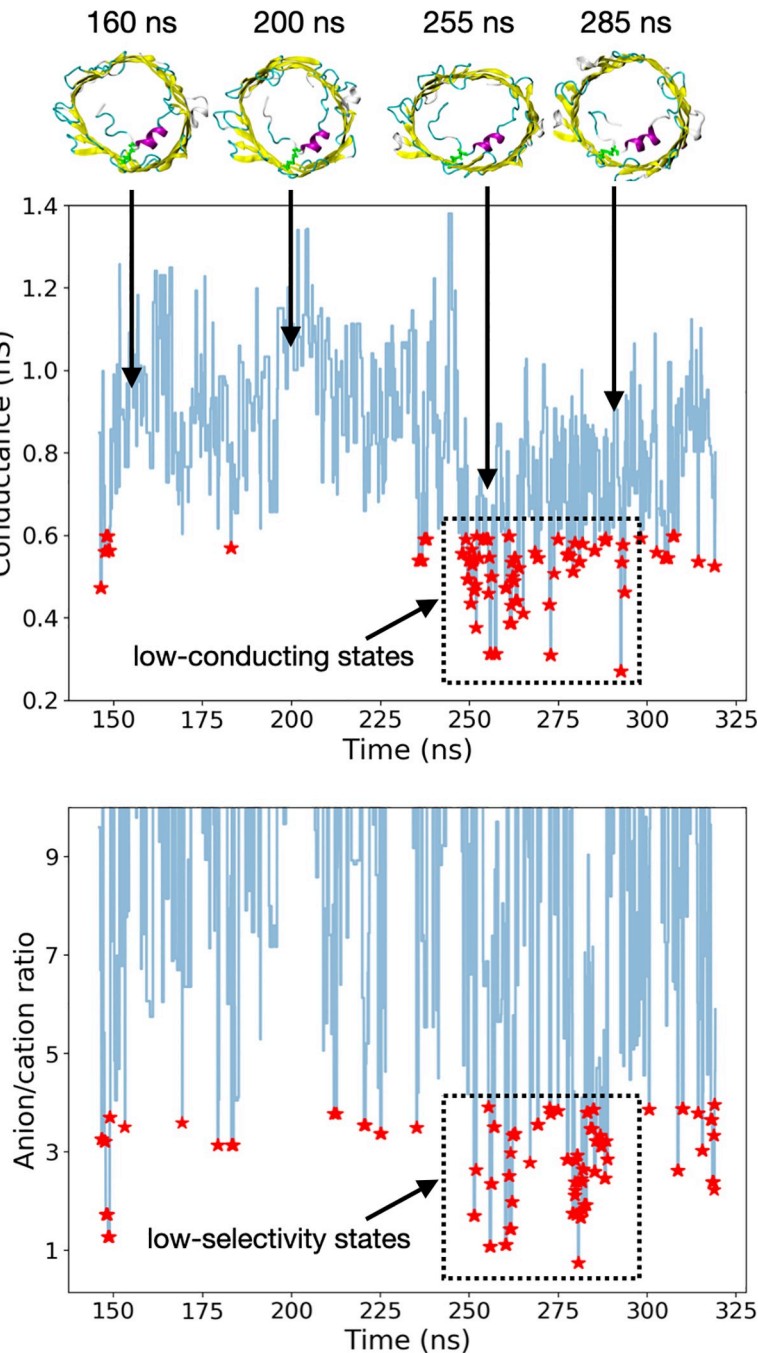

**Fig 5. Conductance and anion/cation ratio of mVDAC1-Cys structures generated at +40mV.** All values were estimated from GCMC/BD simulations at +40mV (150 mM KCl) performed on 500 representative structures of our +40mV aMD trajectory (once the N-terminal tail has detached and moved inside the barrel (~140 ns)). Time series curves were reconstructed by assigning the same conductance and anion/cation ratio to all the frames of a given cluster. Red stars correspond to all the frames with a conductance less than 0.6 nS or an anion/cation ratio less than 4.

cation ratio in S1 was also in line with our GCMC/BD simulations (2.7 at 450 ns) indicating a higher preference for cations although no full cation selectivity was observed (the ratio was always greater than 1).

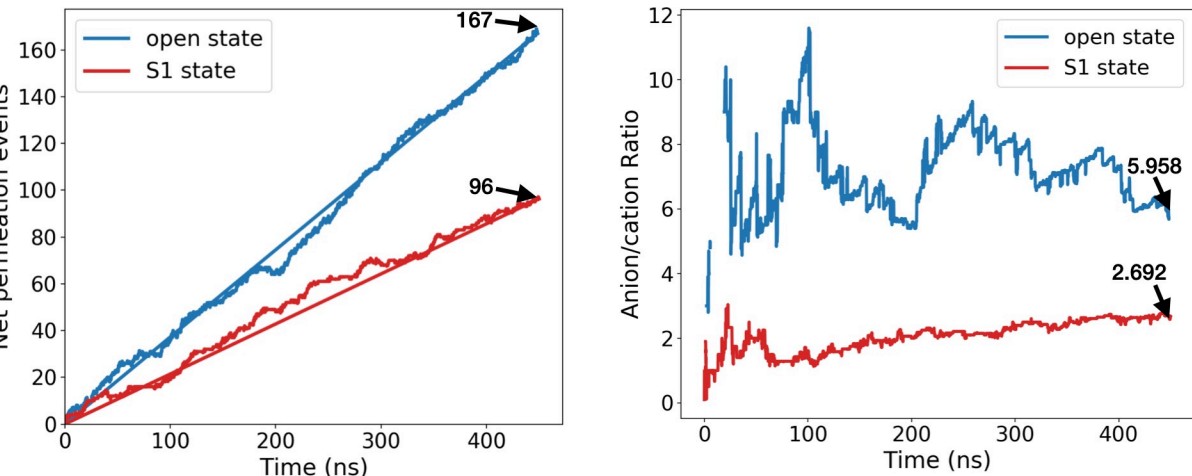

**Fig 6. Net permeation events and anion/cation ratio in mVDCA1-Cys open state and subconducting S1 state.** Values were obtained from 450-ns-long standard MD trajectories at +40mV. In the figure showing net permeation events, the line connecting the first and last points was plotted, the slope of which can be used to estimate the electric current (production MD (channel) in material and methods).

Finally, we explored the structural origin of the low conductance and anion/cation ratio in the S1 state. Note that our 450-ns-long MD trajectories of the open and S1 states can be seen from S6 and S7 Videos, respectively. Interestingly, the trajectory of the S1 state revealed a flexible N-terminal tail (M1 to P5) with different positions and orientations: e.g., the segment was initially pointing toward the cytosolic side while it was more oriented toward the IMS at the end of the trajectory. As the electric current was found to be stable until the end of the simulation (Fig 6), this suggests that residues M1 to P5 are unlikely to fully account for the low conductance. Alternatively, subsequent residues (T6 to D9) always displayed a stable unfolded horizontal position (see also S8 Fig) interacting with the β-barrel via hydrophobic contacts whereas the rest of the N-terminus (C10 to G25) was structurally similar to that of the open state (Fig 7A). Involvement of residues T6 to D9 in the electric current flow was also visible from the density profile of chloride permeation events. This density profile is shown in Fig 7B and 7C as a cross section made of all the (x, y) positions of Cl⁻ ions reaching the middle of the channel. From these figures, we noticed that residues of the unfolded N-terminal segment obstructed the pore in a steric way in the S1 state as illustrated by the complete disappearance of an anion density peak located close to the $3_{10}$-helix in the open state.

Regarding anion selectivity: since the N-terminus was found to block an anion-enriched region within the pore in S1, it was reasonable to observe a decrease in the anion/cation ratio as well. However, our subconducting trajectory also revealed an increase in cation crossing events with respect to the open state (Fig 7D and 7E). In both states, cations were mostly localized close to the barrel wall opposite to the long helix of the N-terminus as was also reported in previous in silico studies [30]. Importantly, this region was found to correlate with the positions of three negatively charged residues, namely, E59, E73 and E84, located next to each other on three consecutive strands (β3, β4 and β5). In S1, the $K^+$ density became higher in this area where we noticed that the side chain of E73 had shifted into the pore while originally pointing toward the membrane (Fig 7E). Notably, E73's reorientation initially occurred in our +40mV aMD trajectory at 70 ns (i.e., before detachment of the N-terminal tail) but was not observed in our aMD runs at 0 mV or -40mV. In this configuration, the side chains of three consecutive residues T72, E73, K74 are facing the pore region which may seem unexpected as consecutive residues in β-sheets usually alternate between each side of the protein backbone.

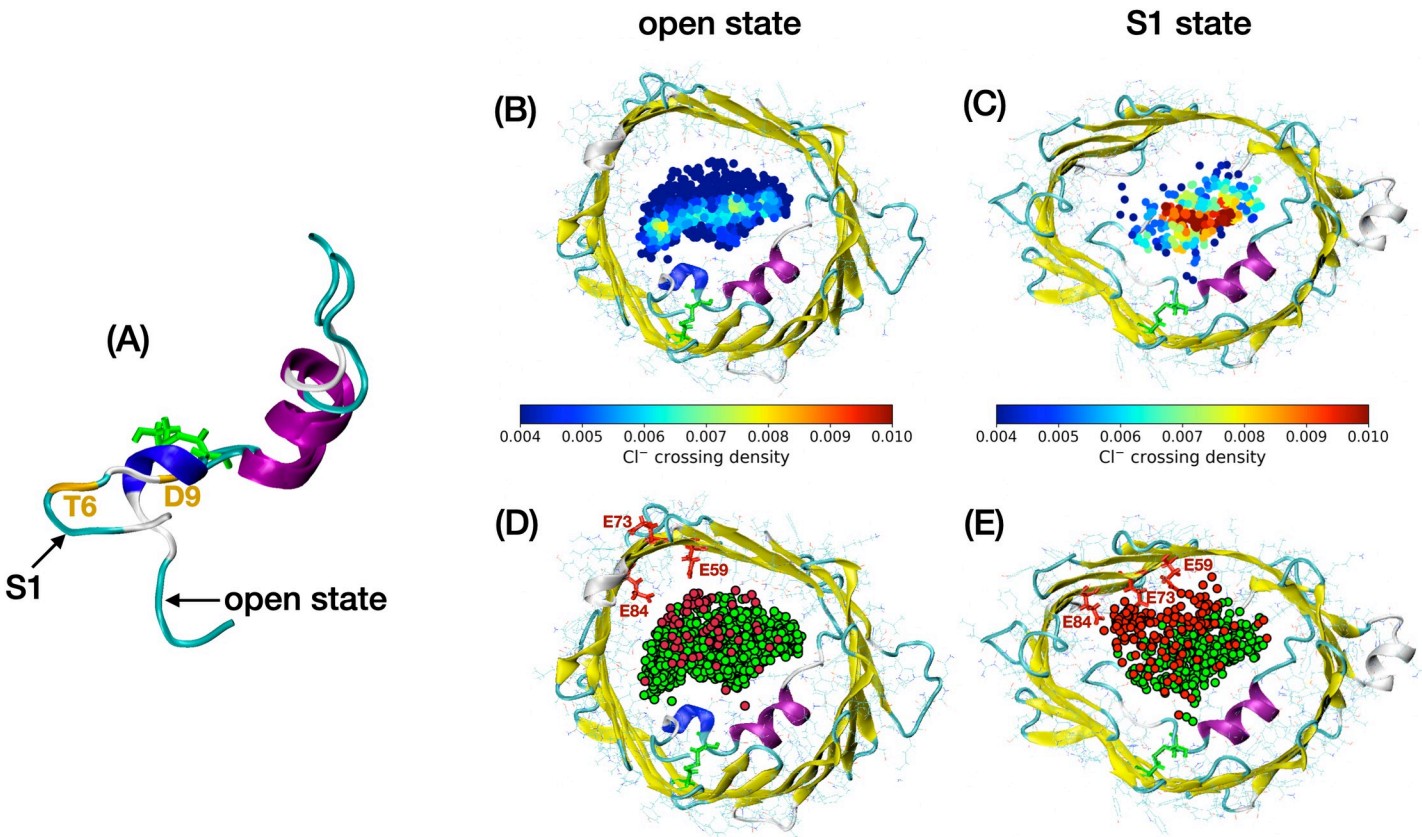

**Fig 7. Open state and subconducting S1 state of mVDCA1-Cys.** (A) side view of the N-terminal region. (B, C): density profile of chloride ions crossing the middle of the channel in the open state (B) and S1 state (C). Crossing events were collected from our 450-ns-long MD trajectories. Note that the density was normalized with respect to the total number of Cl⁻ crossing events in each case (which is about twice as small as in S1). (D, E): positions of potassium ions (in red) and chloride ions (in green) crossing the middle of the channel in the open state (D) and S1 state (E). The open and S1 conformations correspond to representative structures extracted from each MD trajectory. Disulfide bridge linked L10C and A170C residues are shown in green while E59, E73 and E84 residues are colored in red.

However, E73's new orientation remained unchanged until the end of our aMD trajectory as well as in subsequent hundreds-ns-long MD simulations at +40mV confirming its stability at this voltage. Although E73 reoriented early on in our aMD trajectory, decrease in anion selectivity was only observed for a few specific conformers (Fig 4 bottom) suggesting that reorientation of E73 was not sufficient to guarantee cation preference.

## Discussion

In the present study, the intrinsically disordered nature of the N-terminus of VDAC as well as its importance in the gating mechanism of the channel were explored. Although not fully acknowledged until now, several arguments in favor of an intrinsically disordered N-terminal segment in VDAC can be put forward: (i) using NMR and CD techniques, De Pinto and coworkers have shown that a free peptide corresponding to such a domain mostly exists as a random coil in solution [15, 16]. As the N-terminus of VDAC is known to translocate out of the pore of the channel in specific environmental conditions [12], it was suggested that N-terminal unstructured properties were essential in the interaction of VDAC with other proteins [16]; (ii) closer look at the sequence of the N-terminal domain reveals that 13 (out of 25) residues belong to the well-known eight disordered-promoting residues, namely, G, A, S, P, R, Q, E and K. Such residues were originally identified from analysis of various X-ray, NMR and CD

data, as being more prevalent in disordered segments [31]. In VDAC, disordered-promoting residues specifically include A8, G11, K12, S13, A14 and R15, six amino acids involved in the α-helix domain of the N-terminus in the open state; (iii) application of disorder-prediction software like s2D [32] or ESpritz [33] predicted that the N-terminus of VDAC1 was essentially disordered although the s2D method also revealed a propensity of residues K12 to T19 to form helices (S9 Fig). In the case of ESpritz, all the N-terminal residues were predicted to be disordered (i.e., they were all classified as "D") irrespective of the choice of the training set (X-ray or DisProt); (iv) a feature of many disordered systems is that they can fold into different structures when binding to other proteins or protein domains [34]. Interfaces which are crucial to stabilize secondary structures were reported to be significantly more hydrophobic than those of globular protein complexes [35]. In the open state of VDAC, the vast majority of interactions between the helical N-terminus and the β-barrel are non-polar [9]. These include the well-known hydrophobic interactions between L10 and V143-L150 that was shown to be highly conserved across species [8, 9].

Along with the above arguments, further evidence of the disordered properties of the N-terminus of VDAC was provided in the present work where MD performed with the IDR-specific ff14IDPSFF force field enabled to reproduce experimental chemical shifts of the free N-terminal peptide while the standard ff14SB force field was unsuccessful in doing so. From a structural point of view, simulations with ff14SB were characterized by a significant amount of helical content that was most likely responsible for the large deviation from experimental values. Importantly, the tendency of popular force fields to overstabilize secondary structures in disordered systems is not new and is due to force field parameters, especially dihedral correction terms, which are normally calibrated from databases of well-folded proteins [36–38]. Such force fields include ff14SB, CHARMM27 and ff99SB which have been extensively used in previous in silico studies on VDAC [16, 39–41]. Conversely, our simulation with the IDR-specific force field showed almost no α-helical content in the free peptide aside from a small α-turn made of D9, L10 and G11. The complete absence of an α-helical segment similar to the one observed in the crystal structures of VDAC indicates that the α-helix pattern specific of the open state is formed via induced folding through interaction with the β-barrel and not via conformational selection as previously presumed [16]. Moreover, the ability of ff14IDPSFF to properly simulate well-folded systems was confirmed by our MD simulations of mVDAC1 in its open state whereby the N-terminus remained stable all along a 650-ns-long trajectory. Aside from ff14IDPSFF, other force fields have been recently developed to account for intrinsic disordered segments in proteins such as the CHARMM36m force field [42] although their level of accuracy may vary depending on the system being studied [37]. Therefore, application of other force fields to VDAC's N-terminus should require preliminary validation, e.g., from a comparison with experimental chemical shifts as we have done in the present work.

In the second part, we investigated whether the disordered properties of the N-terminus were of importance for the gating mechanism of VDAC. This was done by running aMD simulations on a double cysteine mutant (mVDAC1-Cys) characterized by a disulfide bridge connecting residues L10C and A170C. As shown by Teijido et al. [14], such a setup enabled to induce closed states of the channel while affixing part of the N-terminus against the barrel, thus restricting the exploration of the conformational landscape. Just like the free peptide, striking differences between the ff14SB and ff14IDPSFF force fields were noticed. Whereas the N-terminus of VDAC with ff14SB was insensitive to a +40 mV voltage (no conformational or structural change observed), aMD simulation with ff14IDPSFF showed that the N-terminal tail (M1 to T6) could detach from the β-barrel subsequent to unfolding of the $3_{10}$-helix (Y7, A8 and D9). The tail was found to move further inside the pore while the long α-helix (K12 to T19) remained folded and stably interacting with the barrel all along our simulation. As the N-

terminal peptide was shown to lose its secondary structure in solution, the stability of the long helix in mVDAC1-Cys may be explained by the presence of the C10-C170 disulfide bond whereby residues K12 to T19 are constrained to interact with the barrel thus preserving helical content in this region. Next, using GCMC/BD simulations performed on representative structures generated using ff14IDPSFF, we managed to identify a large number of conformers characterized by a low conductance (~50–60% or below of the conductance in the open state).

Stability of our subconducting conformers was further investigated by running hundreds-ns-long standard MD trajectories using each structure as a starting point. From our analysis, we identified five frames showing long-lived subconducting behavior together with a reduced anion/cation ratio. The five frames were found to be divided into two different structural states, namely, S1 and S2, characterized by different arrangements of the unfolded M1-C10 segment although, in both cases, the segment was interacting with the barrel midway inside the pore. Remarkably, our stable subconducting structures were all reported to match either the S1 or S2 states suggesting that the low conductance was mainly due to the specific positioning of the N-terminus even though such arrangements may have a structural impact on the rest of the channel. For example, we observed that the S2 state exhibited slightly higher ellipticity than S1 or the open state (mean ellipticity values were estimated at 0.182, 0.165, 0.143 from S2, S1 and open states trajectories, respectively). While more thorough investigation might be needed to explore whether the ellipticity is partly responsible for the low conductance, the observed increase in S2 is still well below that of semicollapsed states reported in [11]. Nevertheless, given the multiplicity of VDAC closed states, we leave open the possibility that other stable closed states with higher ellipticity could exist.

Aside from ellipticity measurements, we also started to explore the structural origin of the low conductance and the higher cation preference in the S1 state. Closer look at Cl⁻ ions crossing the middle of the pore indicated that the unfolded T6-D9 segment of the N-terminus further reduced the opening of the pore as compared with the open state. The T6-D9 segment remained stable mostly because of hydrophobic contacts with the barrel as shown in S10 Fig. Interestingly, an increase in cation crossing events was also reported in S1 which contributed to lowering the anion/cation ratio. We observed that $K^+$ ions were mostly concentrated close to a negatively charged motif made of residues E59, E73 and E84 where E73's side chain was found to face the pore region. From our analysis, we were, however, unable to determine the exact involvment of E73 in cation permeation or whether E73's reorientation was favored by our membrane potential. Noticeably, a stable salt bridge involving E84 and K115 was found in the S1 state (S10 Fig) which could also contribute to stabilizing cation-friendly pathways. Regarding cation selectivity, which seems to be typical of closed states in VDAC1 WT [5, 6], none of S1 or S2 were reported to have a anion/cation ratio less than 1. This result may be due to the presence of the disulfide bond which prevents further rearrangement of the N-terminus required for a full cation selectivity [40]. For the record, mVDAC1-Cys was found to exhibit typical gating profile in terms of conductance [14], yet no measurement of the anion/cation ratio was reported that could confirm cation preference of closed states in that case.

Finally, although S1 and S2 were generated from aMD at +40mV, we cannot rule out the possibility that such states may take place through random voltage-independent events. To investigate the specificity of S1 at positive voltage, five 200-ns-long additional trajectories were run at 0mV, +40mV and -40mV. In each trajectory, the RMSD of the T6-D9 segment with respect to our S1 state was computed since T6-D9 was observed to partially block the chloride flux in S1. At +40 mV, 1 trajectory out of 5 was reported to reach conformers similar to S1 (RMSD < 3.0 Å) whereas no such trajectories were found at 0 mV or -40 mV (RMSD > 4.0 Å). Even though our results might indicate that S1 is typical of a positive voltage, further analysis is required to confirm the real impact of the membrane potential. Another possible scenario

being that a high enough voltage, either positive or negative, would be especially important to make the N-terminus unfold (even partially) or detach from the barrel, and that transitions to the multiple closed states would arise in a more random way given the higher flexibility of the unfolded N-terminus. In that case, structurally-similar subconducting states may emerge regardless of the positivity of the voltage. Validation of one model over another is beyond the scope of the present study whose primary objective was to point out that metastable subconducting states of VDAC1, originating from the disordered properties of its N-terminus, can prevail at realistic voltage values.

## Material and methods

### MD software and force fields

All our MD simulations, including standard and aMD simulations, were performed using the Amber16 package [43]. The ff14SB force field [17]. which is known as the latest Amber force field benchmark for proteins simulations is directly available from the Amber package while the ff14IDPSFF force field [18] was obtained from direct request to Prof. Ray Luo's group. ff14IDPSFF can also be downloaded from the Precise Force Field and Bioinformatics Laboratory website: http://cbb.sjtu.edu.cn/~hfchen/index_en.php.

### Structure preparation (peptide)

The N-terminal peptide was obtained by extracting the N-terminal region (M1-G25) from the mVDAC1 structure in the open state (PDB ID: 3EMN [9]). Protonation of the peptide was done using the PropKa algorithm from the MOE 2018 software at a neutral pH [44]. Using the Amber's LEaP module, the system was set into an octahedral box filled with TIP3P waters requesting at least a 15 Å distance between any atom of the peptide and the box edges. Overall, 8826 water molecules were incorporated into the simulation box. Sodium and chloride ions were added to neutralize the system as well as to approximate a 0.01M concentration consistent with experimental conditions used for NMR chemical shifts measurements [16]. Both ff14SB and ff14IDPSFF simulations of the N-terminal peptide were based on the above preparation protocol. However, setting up ff14IDPSFF simulations further required to generate a new topology file containing the proper force field parameters (dihedral correction terms). This was done by running the Trans_FF_IDPs.pl script available with the ff14IDPSFF force field using, as an input, the ff14SB topology file generated from the LEaP module [43]. For each force field, equilibration of the solvated structure was performed with the Amber's pmemd utility by first minimizing the whole system (peptide and solvent) during 5000 steps using the steepest descent method followed by 5000 steps of conjugate gradient. Next, simulations in the NVT ensemble were carried out for 500 ps by slowly heating the system from 0K to 298K while restraining heavy atoms of the peptide. Lastly, NPT simulations at $P$ = 1 bar and $T$ = 298K were run without restraints during 500 ps to finalize the equilibration process.

### Production MD (peptide)

Unbiased MD production was conducted during 1.6 μs and 2.1 μs using the ff14SB and ff14IDPSFF force fields, respectively. aMD simulations performed with the ff14SB force field were run during 1 μs. Details regarding the aMD method and our protocol are given below. Additional aMD simulations starting from a fully unfolded structure of the peptide were run using the ff14SB force field during 20 ns, which was found to be enough to reach the α-helix basin. In that case, the initial conformation was simply extracted from our ff14IDPSFF simulation by selecting the structure with maximum radius of gyration ($\geq$ 20 Å).

## Accelerated MD

To efficiently sample the conformational space of the N-terminal peptide and of mVDAC1-Cys, aMD simulations were run. Briefly, aMD is based on adding a non-negative biasing potential (boost) to the system whenever the system potential is less than an energy threshold [45]

$$V'(\boldsymbol{x}) = V(\boldsymbol{x}) \text{ when } V(\boldsymbol{x}) \geq E$$

$$V'(\boldsymbol{x}) = V(\boldsymbol{x}) + \Delta V(\boldsymbol{x}) \text{ when } V(\boldsymbol{x}) < E, \tag{1}$$

where $V(\boldsymbol{x})$ is the true MD potential, $\Delta V(\boldsymbol{x})$ is the aMD boost and $E$ is the energy threshold. The addition of the $\Delta V(\boldsymbol{x})$ term is done in such a way as to reduce the height of local energy barriers, and is therefore dependent on the energy of the system at $\boldsymbol{x}$. In the present work, aMD was run in its dual boost version, i.e., the aMD boost is the sum of two biasing contributions: one acting on all atoms (including solvent atoms) through the total potential energy $V(\boldsymbol{x})$, and the other one acting only on dihedral angles through the dihedral energy $V_d(\boldsymbol{x})$ such that:

$$\Delta V(\boldsymbol{x}) = \frac{(E_p - V(\boldsymbol{x}))^2}{\alpha_p + E_p - V(\boldsymbol{x})} + \frac{(E_d - V_d(\boldsymbol{x}))^2}{\alpha_d + E_d - V_d(\boldsymbol{x})} \tag{2}$$

Here $E_p$ and $E_d$ correspond to the average (total) potential and dihedral energies of the biased simulations, respectively, whereas $\alpha_p$ and $\alpha_d$ are coefficients accounting for the contribution of each biasing term. Following standard aMD protocols [22, 46], the above coefficients were computed from the following formulas:

$$E_p = E_{p,0} + \gamma_p \cdot n_{atoms}, \alpha_p = \gamma_p \cdot n_{atoms}$$

$$E_d = E_{d,0} + \gamma_d \cdot n_{residues} + \gamma'_d \cdot n_{lipids}, \alpha_d = 0.2(\gamma_d \cdot n_{residues} + \gamma'_d \cdot n_{lipids}), \tag{3}$$

where $\gamma_p = 0.16$ kcal/mol/atom, $\gamma_d = 3.5$ kcal/mol/residue and $\gamma'_d = 30.0$ kcal/mol/lipid. $n_{atoms}$, $n_{residues}$ and $n_{lipids}$ are the number of atoms (including solvent atoms), the number of residues (25 in the case of the peptide) and the number of lipids, respectively. $E_{p,0}$ and $E_{d,0}$ are the average potential and dihedral energies of the unbiased system, respectively, that were estimated from direct analysis of our equilibration simulations in the NPT ensemble.

## aMD reweighting

An important feature of aMD is the possibility to recover ensemble averages associated with the true MD profile. As is well known, structures generated from a biased (equilibrated) simulation characterized by a biasing potential $\Delta V(\boldsymbol{x})$ can be assigned a weight w$(\boldsymbol{x})$ accounting for how they really contribute to the unbiased free energy. This weight is given by the Boltzmann factor: w$(\boldsymbol{x}) = e^{\Delta V(\boldsymbol{x})/k_b T}$ that enables to compute the average of any collective variable $\xi(\boldsymbol{x})$ as

$$\langle \xi \rangle = \frac{1}{\sum_i w(\boldsymbol{x}_i)} \sum_i w(\boldsymbol{x}_i) \xi(\boldsymbol{x}_i) \tag{4}$$

where $\boldsymbol{x}_i$ are the coordinates of the structures generated through the biased simulation.

Notably, the exponential factors can be expressed as a Maclaurin series [47] such that:

$$\mathrm{w}(\boldsymbol{x}) = \sum_{k=0}^{\infty} \frac{1}{k!} \left( \Delta V(\boldsymbol{x})/k_b T \right)^k \tag{5}$$

In practice, finite sampling leads to small errors in the distribution of the $\Delta V(\boldsymbol{x_i})$ while high-order terms in the above series contribute to increase that noise. Therefore, it is of common practice to approximate the weights $\mathrm{w}(\boldsymbol{x})$ by keeping only low-order terms. In this work, all averages quantities deduced from our aMD simulations, including chemical shift estimations, were computed from Eqs (4) and (5) by keeping only terms up to $k = 10$. Although widely accepted, we have tested the applicability of the mentioned formulas on a toy model – alanine dipeptide in explicit solvent – by confirming that averages of $\varphi$, $\psi$ dihedral angles of the peptide from aMD data could well reproduce averages obtained from standard MD.

## Chemical shifts

Cα and Cβ chemical shifts were calculated using the SPARTA+ software [20] which relies on a neural network algorithm to predict the shifts from an input protein structure. The high execution speed of the program enabled to compute the chemical shifts for each structure saved throughout our MD or aMD simulations (one structure saved every 2 ns). Standard averages of each chemical shift – or weighted averages in the case of aMD data – were computed from those predictions and directly compared with experimental data (Table S1 in Ref. [16]).

## Structure preparation (channel)

Structures of mVDAC1 and mVDAC1-Cys were prepared from the 3EMN crystal structure by aligning the protein to its corresponding *Orientation of Protein in Membranes* (OPM) structure [48]. This enabled to translate the system so that its center of mass was positioned at the origin and the pore was oriented along the z axis. In the case of mVDAC1-Cys, the MOE program [44] was used to mutate residues L10 and A170 into cysteines and to create the disulfide bond. The channel was then embedded into a DOPC/DOPE membrane using the CHARMM-GUI membrane builder webserver [49]. A rectangular simulation box with TIP3P water was considered. The membrane was built following a 2:1 PC:PE ratio as it was shown to be representative of the lipid composition of the MOM [50]. Lipids were positioned in such a way as to roughly cover a 90 x 90 Å$^2$ surface in x and y directions which resulted in 186 lipid molecules. Thickness of water layers on both sides of the membrane was set to 22.5 Å creating around 12500 water molecules in the box. Protonation of protein residues was performed with the membrane builder tool. As suggested in Ref. [41], the E73 residue was kept in its unprotonated form although it was initially facing the membrane. A 150-mM KCl concentration was applied in all our simulations. Finally, equilibration protocol involving minimization, NVT and NPT steps, was similar to the one used for the free peptide except NPT equilibration was run during 5 ns (mVDAC1) or 10 ns (mVDAC1-Cys) using an anisotropic pressure scaling as is required for membrane systems. See "structure preparation (peptide)" for other parameter values.

## Production MD (channel)

All production runs of wild-type mVDAC1 were performed in the NPT ensemble while those related to mVDAC1-Cys were conducted in NVT starting from structures equilibrated in NPT. All our runs were simulated at 150-mM KCl concentration. A summary of our exploratory MD and aMD simulations is provided in S2 Table. Transmembrane (TM) potential was modeled by applying an external electric field along the z-axis using Amber's "efz" parameter.

Voltages include 0 mV, +40 mV and −40 mV. The reason for running production in NVT was due to the inability of the Amber package to account for virial contribution in NPT upon application of an external field. To investigate the conductance and the stability of mVDAC1-Cys' open state as well as of the 51 subconducting conformers generated from aMD, 200-ns-long standard MD trajectories were run at +40mV using each structure as a starting point (one trajectory per conformer). The starting structure in the open state corresponds to our initial structure after 10 ns equilibration in NPT (see "structure preparation (channel)"). To estimate the electric current, linear regression of the net ion crossing events as a function of time was performed. Since the diffusion coefficient of ions is usually overestimated in MD [29], we first computed the bulk diffusion coefficient of potassium chloride at 10 mM as predicted from standard simulation in TIP3P water. A 750-ns-long trajectory involving 8 $K^+$ and 8 $Cl^-$ ions in a box containing around 36700 water molecules resulted in a KCl diffusion coefficient $D_{sim}$ = 3.273. $10^{-5} cm^2$/s ($R^2$ = 0.99) whereas the real diffusion coefficient is known to be $D_{exp}$ = 1.918. $10^{-5} cm^2$/s at the same KCl concentration [51]. The $D_{exp}/D_{sim}$ ratio was then used as a multiplicative factor to estimate the current from the slope of the *crossing-events-vs-time* curve.

## GCMC/BD

Poisson-Boltzmann and Poisson-Nernst-Planck (PB/PNP) and GCMC/BD calculations were run using the open-source PB/PNP and GCMC/BD programs available at http://www.charmm-gui.org/?doc=input/gcmcbd. For more details about the PB/PNP and GCMC/BD protocols, see Refs. [27, 28]. GCMC/BD simulations were carried out on representative frames of our aMD trajectories of mVDAC1-Cys at 0 mV, +40 mV and −40 mV. RMSD clustering analysis was performed from the full trajectory generated at 0 mV, from 140 ns at +40 mV, i.e., when the N-terminal tail has moved inside the barrel, and from 300 ns at −40 mV. In all cases, 500 clusters resulting in 500 frames were generated. Next, the CHARMM-GUI Ion Simulator webserver [52] was used to generate all the input files needed for PB/PNP and GCMC/BD calculations. Each structure was embedded into a 35-Å-thick membrane oriented along the z-axis and positioned in the middle of an 85 x 85 x 95 $Å^3$ box. Thickness of buffer regions at the top and bottom of the simulation box were set to 5 Å while a fixed concentration of 150mM KCl was used on both sides of the membrane. PB/PNP was carried out prior to GCMC/BD in order to estimate the protein electrostatic and steric potentials at each point of a rectangular grid with a grid spacing of 0.5 Å. These values are used in GCMC/BD simulations. Each GCMC/BD run was performed by conducting one GCMC step every BD step for a total of $5 \times 10^7$ BD cycles. Values reported in Figs 5 and S6 were obtained by averaging over 3 independent GCMC/BD replicas. The TM voltage was set to +40 mV for frames extracted from aMD at 0 mV and +40 mV, and to −40 mV for frames extracted from aMD at −40 mV. Conductance was estimated by summing up K+ and Cl- currents and dividing by the applied voltage. Anion/cation ratio was computed as the ratio of the Cl- flux to the K+ flux.

## Supporting information

**S1 Fig. Convergence of standard MD simulations of the VDAC1 N-terminal peptide.** The figure shows the number of clusters generated as a function of the simulation time in the case of the ff14SB force field (A) and of the ff14IDPSFF force field (B). Clustering analysis was performed using a hierarchical agglomerative approach with an RMSD threshold of 3 Å. (TIF)

**S2 Fig. PMF profile generated from standard MD simulations of the VDAC1 N-terminal peptide.** PMFs were obtained in the cases of ff14SB (top) and ff14IDPSFF (bottom) and are

given as a function of the radius of gyration ($R_g$) and the RMSD with respect to the N-terminal peptide conformation in the 3EMN crystal structure. Representative structures in main basins and transition regions are also depicted. The top figure was built out of a 1.6 μs-long trajectory while data at the bottom were collected over 2.1 μs.
(TIF)

**S3 Fig. PMF profile generated from aMD simulation of the VDAC1 N-terminal peptide with the ff14SB force field.** Due the biased nature of the simulation, density values were reweighted from a Maclaurin series up using terms up to rank $k = 10$ (Eq (5)). As in S2 Fig, the PMF is given as a function of the radius of gyration ($R_g$) and the RMSD with respect to the N-terminal peptide conformation in the crystal structure of mVDAC1. The dashed dark-blue trajectory represents a 20-ns-long aMD trajectory generated by initiating our simulation from a fully unfolded structure (dark blue cross) that clearly shows the convergence to the helix-enriched basin.
(TIF)

**S4 Fig. Secondary structure content of wild-type mVDAC1 N-terminus.** Values were obtained from 100-ns-long MD runs using the ff14SB force field (left) and the ff14IDPSFF force field (right). Note that β-content is not displayed as it is everywhere zero and helical content includes both $3_{10}$-helical and regular α-helical content. For both force fields, $3_{10}$-helical content is found only in the short helix made of residues Y7 to D9 and regular helix content was only observed in the long helix made of residues K12 to T19.
(TIF)

**S5 Fig. Representative conformers of mVDAC1-Cys at 0mV, +40mV and -40 mV.** Conformers were obtained from agglomerative hierarchical clustering analysis of aMD trajectories using the RMSD of the N-terminus as a metric and a threshold distance of 3.0 Å. The representative structure (viewed from the IMS) of the 5 most populated clusters is shown at each voltage. Disulfide bridge linked L10C and A170C residues are shown in green.
(TIF)

**S6 Fig. Conductance and anion/cation ratio of mVDAC1-Cys structures generated from aMD at 0mV, +40 mV and -40mV.** All values were obtained from GCMC/BD runs at 150-mM KCl concentration performed on 500 representative structures of each aMD trajectory. Time series curves were reconstructed by assigning the same conductance and anion/cation ratio to all the frames of a given cluster. Red stars correspond to all the frames with a conductance less than 0.6 nS or an anion/cation ratio less than 4. Note that the frames extracted from our aMD trajectories at +40mV and -40mV were selected from 140 ns and 300 ns, respectively.
(TIF)

**S7 Fig. Subconducting mVDAC1-Cys structures identified from standard MD.** The frames (IMS view) correspond to the rows highlighted in green in S3 Table, namely frames 4, 12, 30, 37 and 46. The first cluster (S1 state) is made of frames 30 and 46 while the second cluster is made of frames 4, 12 and 37 (S2 state).
(TIF)

**S8 Fig. RMSD in mVDAC1-Cys open state and subconducting S1 state.** The backbone RMSD of the whole channel as well as the RMSD of the T6-G25 segment are shown.
(TIF)

**S9 Fig. Application of the sequence-based protein secondary structure prediction s2D method on the VDAC1 N-terminus.** The graph displays random-coil and secondary structure propensity for each residue.
(TIF)

**S10 Fig. Hydrogen bonds and hydrophobic contacts in mVDAC1-Cys open and S1 states.** Each graph was built out of 25 frames equally spaced in time in each 450-ns-long trajectory. Only contacts occurring in 50% of cases or more are depicted. Red squares correspond to hydrogen bonds while blue squares stand for hydrophobic contacts. Note that contacts are only displayed in the upper left corner of each graph to avoid redundancy. In S1 state, the dashed rectangle includes hydrophobic contacts that stabilize the N-terminus against the barrel wall. Other interactions such as the E84-K115 salt bridge are shown.
(TIF)

**S1 Table. Root-mean-square error (RMSE) and Pearson correlation coefficient (R) obtained by comparing experimental and simulated chemical shifts of the VDAC1 N-terminal peptide.** Simulations include MD and aMD ones using either the ff14SB or the ff14IDPSFF force field.
(TIF)

**S2 Table. Molecular simulations of mVDAC1 (WT and double Cys mutant) performed in the present study.**
(TIF)

**S3 Table. Current, conductance and anion/cation ratio of mVDAC1-cys frames selected from aMD at +40mV.** The values were predicted by running 200-ns-long standard MD for each frame and by recording ion permeation events. The $R^2$ coefficient obtained from linear regression of the *crossing-events-vs-time* curve is also displayed. The current was computed by multiplying the predicted slope of the curve by a scaling factor (see production MD (channel) in material and methods). Rows highlighted in green correspond to frames displaying both reduced conductance (around 0.6 nS or less) and a low anion/cation ratio (around 4 or less). Frames 30 and 46 correspond to the same structural S1 state while frames 4, 12 and 37 belong to S2 state (S7 Fig). Rows highlighted in orange stand for frames showing only low anion/cation ratio while rows in grey-blue are related to subconducting states with high anion/cation ratio (greater than 4.5).
(TIF)

**S1 Video. Clip of the MD trajectory of wild-type mVDAC1 (650 ns) using ff14IDPSFF in the absence of applied voltage.** The channel is viewed from the expected cytoplasmic side of the membrane. No conformational transition or change in secondary structure content was observed.
(MP4)

**S2 Video. aMD trajectory of mVDAC1-Cys (420 ns) at +40mV using ff14SB.** The channel is viewed from the cytoplasm. Disulfide bridge linked L10C and A170C residues are shown in green. No conformational transition or change in secondary structure content was observed.
(MP4)

**S3 Video. aMD trajectory of mVDAC1-Cys (540 ns) at 0mV using ff14IDPSFF.** The channel is viewed from the cytoplasm. Disulfide bridge linked L10C and A170C residues are shown in green. The $3_{10}$-helix (in dark blue) made of residues Y7, A8 and D9 was found to unfold at 160 ns. However, the position of the N-terminal tail (M1-P5) and of the long helix (K12 to

T19) remained unchanged.
(MP4)

**S4 Video. aMD trajectory of mVDAC1-Cys (320 ns) at +40mV using ff14IDPSFF.** The channel is viewed from the cytoplasm. Disulfide bridge linked L10C and A170C residues are shown in green. The $3_{10}$-helix (in dark blue) made of residues Y7, A8 and D9 was found to unfold at 20 ns. Subsequently, the N-terminal tail (M1-P5) moves further inside the barrel (at 140 ns) leading to steric hindrance of the pore.
(MP4)

**S5 Video. aMD trajectory of mVDAC1-Cys (595 ns) at -40mV using ff14IDPSFF.** The channel is viewed from the cytoplasm. Disulfide bridge linked L10C and A170C residues are shown in green. The $3_{10}$-helix (in dark blue) made of residues Y7, A8 and D9 was found to unfold at 2 ns.
(MP4)

**S6 Video. MD trajectory of mVDAC1-Cys in the open state (450 ns) at +40mV using the ff14IDPSFF force field.** The channel is viewed from the cytoplasm. Disulfide bridge linked L10C and A170C residues are shown in green.
(MP4)

**S7 Video. MD trajectory of mVDAC1-Cys in the S1 state (450 ns) at +40mV using the ff14IDPSFF force field.** The channel is viewed from the cytoplasm. Disulfide bridge linked L10C and A170C residues are shown in green.
(MP4)

## Acknowledgments

We gratefully acknowledge support from the PSMN (Pôle Scientifique de Modélisation Numérique) of the ENS de Lyon for computing resources.

## Author Contributions

**Conceptualization:** Jordane Preto, Isabelle Krimm.

**Formal analysis:** Jordane Preto.

**Investigation:** Jordane Preto.

**Methodology:** Jordane Preto, Isabelle Krimm.

**Resources:** Isabelle Krimm.

**Software:** Jordane Preto.

**Supervision:** Isabelle Krimm.

**Visualization:** Jordane Preto.

**Writing – original draft:** Jordane Preto, Isabelle Krimm.

**Writing – review & editing:** Jordane Preto, Isabelle Krimm.

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
