## [Decision Letter · Decision Letter 0]

19 Oct 2020

Dear Dr. Preto,

Thank you very much for submitting your manuscript "The intrinsically disordered N-terminus of the voltage-dependent anion channel" for consideration at PLOS Computational Biology.

As with all papers reviewed by the journal, your manuscript was reviewed by members of the editorial board and by several independent reviewers. In light of the reviews (below this email), we would like to invite the resubmission of a significantly-revised version that takes into account the reviewers' comments.

We cannot make any decision about publication until we have seen the revised manuscript and your response to the reviewers' comments. Your revised manuscript is also likely to be sent to reviewers for further evaluation.

Sincerely,

Alexander MacKerell

Associate Editor

PLOS Computational Biology

Arne Elofsson

Deputy Editor

PLOS Computational Biology

Reviewer's Responses to Questions

**Comments to the Authors:**

Reviewer #1: No comments

Reviewer #2: The paper describes molecular dynamics (MD) simulations of mVDAC beta-barrel channel protein focusing on conformational transitions of N-terminal peptide, which seems to play a crucial role in the channel gating. The authors demonstrated that AMBER force field ff14IDPSFF, which explicitly considers intrinsically disordered regions (IDRs) in protein structure, provides much more accurate description of the N terminal peptide in aqueous solution using experimental chemical shift data compared to a standard AMBER protein force field   They also demonstrated using accelerated MD (aMD) simulations possible open to closed channel transitions for applied voltage simulations of double Cys  L10C-A170C mVDAC mutant, which anchors a part of N terminal domain to a beta barrel via disulfide bond. The flexible N terminal end unwinds upon applied +40 or -40 mV voltage and can partially block the pore decreasing conduction and anion/cation selectivity of the channel.

The paper is written well, figures and movies are also of high-quality and show a nice depiction of key points from the study. The study is of significance and demonstrates  a propensity of N-terminus of VDAC to favor an intrinsically disordered state  and a new interesting look at VDAC gating (but see below).

However, there are some concerns regarding interpretation of  L10C-A170C mVDAC simulations related to channel gating. It does not seem that the authors demonstrated consistent gating transition toward the closed state of the channel, which may happen on millisecond or even longer time scales (depending on the applied voltage and other factors).  The authors observed partial unfolding of an untethered N terminal portion of the peptide under an applied voltage, which seems to lead to lower conductance and reduced selectivity based on GCMC/BD simulations they performed using all-atom aMD structures. It seems that these transitions may rather correspond to transient subconductance levels also mentioned in a previous study they cited (Briones et al  Biophys J. 2016 Sep 20; 111(6): 1223–1234.).

Moreover, the authors did not seem to see (and/or analyzed) consistent changes in the rest of the voltage sensor, crucial for voltage-dependent transitions, as suggested in the experimental study they used as a basis of their simulations (Teijido et al J Biol Chem. 2012 Mar 30; 287(14): 11437–11445.):

"During voltage-induced gating, the N-terminal region either stays inside the pore or moves in a concerted fashion with the rest of the voltage sensor rather than exerting large independent structural rearrangements to achieve voltage-dependent gating. It should be noted that the data presented here still permit the translocation of the N terminus in and out of the lumen upon gating, as suggested previously (16, 20, 32, 46), as long as this movement is coordinated with the voltage sensor."

The structural changes the authors observed occurred on the time scale of only few to few hundred ns, many orders of magnitude shorter than physiological time scales for gating transitions.  Although, effective time scales may be somewhat longer in aMD runs.  

Since only singular, albeit pretty long aMD simulations were performed, there is also a possibility that presented trends may represent just random events rather than voltage-dependent changes. And this may be suggested by quite different outcomes of +40 and -40 mV simulations.Also, typically in MD simulations an applied voltage needs to be increased compared to physiological values to sample slow gating transitions during MD simulation time frame. So, using small physiological values is somewhat encouraging but again leaves a possibility of random voltage-independent events.

Also, related to this - how TM voltage was applied in simulations is not discussed.

It should be also noted that even through CHARMM22+CMAP protein force field used in some of previous VDAC MD studies does not account explicitly for protein IDRs , this is done in the newest force field version - CHARMM36m  (Huang et al Nat Methods  2017 Jan;14(1):71-73).

There are also quite a few specific suggestions and corrections:

p 7   It may help to make experimental data as thick lines in Fig 2

p 8 why  aMD was not used for ff14IDPSFF simulations?

p 9  "from  S1 Table" -> "from  Table S1"

p 10 l 208 "within the time frame of our standard MD (650 ns)" - However, 150 ns long MD simulation was mentioned earlier

p 13 Fig 4. It is not clear whether structures shown represent a snapshot at a particular simulation time (e.g. 10 ns) or some representative structures from the whole simulation. Also, it will be helpful if panels are labeled A and B or A, B, C and D. In the left panel, it will be helpful to show structures from different simulations by different colors or representations.  

p 14 l 279 Why 0 mV MD simulations were not used for GCMC/BD runs at -40 mV, but used for runs at 40 mV.

p 14 l 283 "given at" ->  "of"

p 15 l 304 "Time curves" -> "Time series curves"

p 16. l 311 "leading to slight compression of the latter in the region of interaction" Where such compression is demonstrated?

p 16 l 322 "Intensity" - current. The same in y axis labels in Fig 6.

p 16 Fig. 6 Right panel legend should be moved to the center not to obstruct one of the curves. Labels A and B would be helpful.

p 17 l 337 "unstructured properties" -> "unstructured conformation"

p 17 l 339 "G, A, S, P, R, Q, E and K" This list from ref. 30 includes amino acids enriched both fibrous and non-fibrous low-complexity proteins, and it's not clear whether the former are relevant for this work.

p 17 l 345 "N-terminus were disordered (not shown)." This along with at least one other  secondary-structure prediction may need to be shown in Sup Info, as this is crucial for the study

p 18 l 374 "made of" -> "creating"

p 18 ll 375 "As shown by Teijido et al. (14), such an experiment enabled to

p 19 l 383 "stable interacting" -> "stably interacting"

p 19 l 391 "few" -> "a few" ?

p 21 l 449 "MD production" -> "production MD"

p 25 l 533  It is not clear where PB/PNP was used

pp 25-26 ll 538-539 "500 clusters resulting in 500 frames were generated" - did each cluster contain just one structure or center of each cluster was selected?

p 28 l 572 S4 fig. caption "wild-type mVDAC1" -> "wild-type mVDAC1 N-terminus"

p 28 ll 578-588 Figs S5 and S6 captions. Not only a disulfide bond, but also bonded Cys residues are shown in green, and H bonding residues seem to be shown in pink as well.

Also, there is no similar figure for +40 mV simulation.

p 29 l 593 "Time curves" -> "Time series curves"

p 29 l 599 Table S1. Is it R or R^2, as more commonly reported? same for Table itself.

p 29 l 602 "Characteristics" -> "Parameters"?

Fig S2 can be a main text Figure

Figs. S5 and S6 - % of two lower conformations are not shown

Also, there is no similar figure for +40 mV simulation.

Fig S7. Contact map is really confusing. I would just focus on the left corners and exclude adjacent residue interactions, which are the same.  And in my opinion time series plots of most important interactions with labeling low-conductance states would be much more informative.  And  0 mV simulation can be added in that case as well.

Throughout the text it is confusing to see e.g. "S1 Figure" or "S2 Table" instead of "Table S1" or "Figure S2"

Reviewer #3: The manuscript by Preto & Krimm deals with the investigation of a voltage dependent anion channel from mithocondria, VDAC-1. This channel is involved in several cellular transport processes, such as ions and small metabolites, contributing to cell homeostasis. Its interest is largely due to its supposed involvement in apoptosis, thus it represents an interesting target for cancer and neurodegenerative diseases.

It is a beta-barrel pore with the N-terminus partially folded inside at middle eight. High-resolution NMR and X-ray structures do exist showing the strong interactions between the barrel and the N-terminus. In particular, it is known from electrophysiology it can gate at voltages larger that 30 mV (absolute), however the mechanism of the voltage closure it is still debated in literature. Its open-state properties were well described in literature, especially using molecular dynamics simulations.

Previous simulations used either the CHARMM of the AMBER force field, the most popular force field. However these force fields has been developed to simulate proteins in their folded states. A previous investigation (ref 16) showed, combining NMR and simulations, how the N-terminal in solution has some aspect typical of a disordered peptide. Here the authors had the brilliant idea to use a different force field, developed for describing intrinsically disordered proteins. The new simulations performed here are in better agreement when comparing the NMR chemical shifts (taken from ref 16) with respect to the use of FF14SB of ref 16.

Then they investigated the gating properties simulating a specific double-mutant system with realistic voltages (really appreciated…). By selecting some representative conformers from biased simulations, they applied GCMC/BD to estimate the conductance. As expected, some of the conformers show low conductance with lower anion selectivity, typical of the closed state.

A few comment to improve the paper:

- From fig. 4 we see a deformation of the barrel, from spherical to elliptical. This is perhaps not due only to the double cysteine mutant, it was already observed in simulation of WT VDAC-1, as described in the paper Amodeo, G. F., et al. PLoS ONE 9, e103879 (2014). The author should comment on it.

- Not sure about their conclusion: “no apparent increase in the ellipticity of the barrel was reported during the formation of our subconducting states…” It seems to me that in fig. 4 the conformers 3 and 4 (from left to right) have a different ellipticity than 1 and 2. See also my previous comment. Perhaps the collapse is only minimal, however it might be induced by the presence of the disordered state of the N-terminal. They accelerated only some collective variables. I would expect that the rearrangement of the barrel toward a more elliptical stable state might occur on time scale not reached by the present simulations.

- It would be interesting to see the path of ions in CGMC/BD simulations, in two representative of both the closed and open states.

**Have all data underlying the figures and results presented in the manuscript been provided?**

Reviewer #1: None

Reviewer #2: Yes

Reviewer #3: Yes

PLOS authors have the option to publish the peer review history of their article (what does this mean?). If published, this will include your full peer review and any attached files.

Reviewer #1: No

Reviewer #2: No

Reviewer #3: No
---

## [Decision Letter · Decision Letter 1]

29 Dec 2020

Dear Dr. Preto,

Thank you very much for submitting your manuscript "The intrinsically disordered N-terminus of the voltage-dependent anion channel" for consideration at PLOS Computational Biology. As with all papers reviewed by the journal, your manuscript was reviewed by members of the editorial board and by several independent reviewers. The reviewers appreciated the attention to an important topic. Based on the reviews, we are likely to accept this manuscript for publication, providing that you modify the manuscript according to the review recommendations.

Sincerely,

Alexander MacKerell

Associate Editor

PLOS Computational Biology

Arne Elofsson

Deputy Editor

PLOS Computational Biology

[LINK]

Reviewer's Responses to Questions

**Comments to the Authors:**

Reviewer #2: The manuscript describes all-atom molecular dynamics (MD) simulations of the murine VDAC1 channel in the open state focusing on the conformation of its N-terminal region, which represents an intrinsically disordered region (IDR). They convincingly demonstrated that with MD simulations of the isolated N-terminus in aqueous solution and showing good agreement with experimental chemical shift values for the C_alpha and C_beta protein atoms when using ff14IDPSFF force field, optimized for protein IDRs. Furthermore, they performed multiple unbiased and accelerated MD (aMD) runs of the entire lipid bilayer embedded VDAC1 with and without applied voltage and demonstrated conformational transitions of the N-terminal regions, leading in some cases to low-conductance and lower anion-selectivity channel conformational states. They performed multiple analyses of those simulations identifying structural determinants of those states.

This is a revised version of the manuscript. The authors performed an outstanding job to address multiple reviewers' comments. They also performed multiple new simulations and analyses to further prove their conclusions, which add a lot of value. I also want to commend the authors for the attention to detail and and overall excellent presentation style. This is a really top-quality manuscript with high significance. I do not have any major comments or suggestions at this point. I have a list of very minor, mostly wording suggestions listed below.

p 4 l 85 "Chloride and potassium ions" - please specify colors

p 5 l 109 "cis side" - please specify more explicitly or omit to avoid confusion as it's discussed further in the text

p 7 "beta-carbons chemical shifts" -> "beta-carbon chemical shifts"

p 8 l 172 "ff14SB, were" -> "ff14SB were" (remove comma)

p 9 l 180 "overexpress" -> "overstabilize"

p 12 l 251 "hydrogen bonds at" -> "hydrogen bonds between"

p 12 l 255 "Bonded" -> "Disulfide bridge linked"

p 12 l 256 "IMS view" Maybe also specify top or bottom relative to structure shown in Fig. 1 and/or specify in Fig. 1, which one is IMS/cytoplasmic side and also maybe cis/trans sides.

p 13 l 260, 261"hydrogen bonds at" -> "hydrogen bonds between"

p 13 l 264 "hydrophobic patch" -> "hydrophobic interaction or contact"

p 19 l 371 "around twice smaller" -> "about twice as small as"

p 12 l 374 "Bonded" -> "Disulfide bridge linked"

p 19 l 419 "hydrophobic patch" -> "hydrophobic interaction or contact"

p 22 l 443 "experimental shifts" -> "experimental chemical shifts"

p 24 l 484 "K+" -> "K^+"

p 25 l 511 "can be prevail at" -> "can prevail at"

p 26 l 543 "MD production" -> "production MD"

p 30 l 616 "150nM KCl" -> "150 mM KCl" ? 150 nM is a very low, not physiological concentration and not possible to model via MD in most cases

p 30 l 628 "at 10nM" -> "at 10 mM"? please see comment above

p 31 l 648 "on each point" -> "at each point"

p 31 l 650 "5. 10^7" -> "5x10^7"

p 32 ll 661, 667 "of the N-terminal peptide." -> "of the VDAC1 N-terminal peptide."

p 32 ll 664, 670 "N-terminal conformation" -> "N-terminal peptide conformation"

p 32 l 678 "regular helix content" -> "regular alpha-helical content"

p 32 l 683 "Bonded" -> "Disulfide bridge linked"

p 32 l 685 Please specify viewpoint e.g. with respect to Fig. 1top/bottom and/or IMS/cytoplasmic

p 33 l 695 same as above

p 33 l 698 "of the s2D method" -> "of the sequence-based protein secondary structure prediction s2D method"

p 33 l 700 "open state and S1" -> "open and S1 states"

p 32 l 707 RMSE -> "Root-mean-square error (RMSE)"

p 34 l 708 "of the N-terminal peptide." -> "of the VDAC1 N-terminal

p 34 ll 725, 728 "Bonded" -> "Disulfide bridge linked"

p 35 ll 732, 736, 740, 742 "Bonded" -> "Disulfide bridge linked"

pp 34 - 35 Movie S1-S7 captions. Please specify viewpoints e.g. with respect to Fig. 1 top/bottom and/or IMS/cytoplasmic. You may also specify color scheme used for alpha-helix, 3-10 helix, beta-barrel etc. or refer to previous figures.

Reviewer #3: The authors replied to all my comments

**Have all data underlying the figures and results presented in the manuscript been provided?**

Reviewer #2: Yes

Reviewer #3: Yes

PLOS authors have the option to publish the peer review history of their article (what does this mean?). If published, this will include your full peer review and any attached files.

Reviewer #2: No

Reviewer #3: No
---

## [Editor Report · Decision Letter 2]

27 Jan 2021

Dear Dr. Preto,

We are pleased to inform you that your manuscript 'The intrinsically disordered N-terminus of the voltage-dependent anion channel' has been provisionally accepted for publication in PLOS Computational Biology.

Best regards,

Alexander MacKerell

Associate Editor

PLOS Computational Biology

Arne Elofsson

Deputy Editor

PLOS Computational Biology

---

## [Editor Report · Acceptance letter]

7 Feb 2021

PCOMPBIOL-D-20-01664R2 

The intrinsically disordered N-terminus of the voltage-dependent anion channel

Dear Dr Preto,

I am pleased to inform you that your manuscript has been formally accepted for publication in PLOS Computational Biology. Your manuscript is now with our production department and you will be notified of the publication date in due course.

With kind regards,

Alice Ellingham
